# Population genomics of Group B Streptococcus reveals the genetics of neonatal disease onset and meningeal invasion

Chrispin Chaguza [1,2,8 ✉], Dorota Jamrozy [1,8], Merijn W. Bijlsma[3,8], Taco W. Kuijpers[4,5], Diederik van de Beek[3], Arie van der Ende [6,7,9 ✉] & Stephen D. Bentley [1,9 ✉]

Group B *Streptococcus* (GBS), or *Streptococcus agalactiae*, is a pathogen that causes preterm births, stillbirths, and acute invasive neonatal disease burden and mortality. Here, we investigate bacterial genetic signatures associated with disease onset time and meningeal tissue infection in acute invasive neonatal GBS disease. We carry out a genome-wide association study (GWAS) of 1,338 GBS isolates from newborns with acute invasive disease; the isolates had been collected annually, for 30 years, through a national bacterial surveillance program in the Netherlands. After controlling for the population structure, we identify genetic variation within noncoding and coding regions, particularly the capsule biosynthesis locus, statistically associated with neonatal GBS disease onset time and meningeal invasion. Our findings highlight the impact of integrating microbial population genomics and clinical pathogen surveillance, and demonstrate the effect of GBS genetics on disease pathogenesis in neonates and infants.

[1] Parasites and Microbes Programme, Wellcome Sanger Institute, Wellcome Genome Campus, Hinxton, Cambridge, UK. [2] Department of Epidemiology of Microbial Diseases, Yale School of Public Health, Yale University, New Haven, CT, USA. [3] Department of Neurology, Amsterdam Neuroscience, Amsterdam University Medical Center, University of Amsterdam, Amsterdam, The Netherlands. [4] Department of Immunopathology, Sanquin Research and Landsteiner Laboratory of the Academic Medical Center, University of Amsterdam, Amsterdam, The Netherlands. [5] Department of Paediatric Haematology, Immunology and Infectious Diseases, Emma Children's Hospital, Amsterdam University Medical Center, Amsterdam, The Netherlands. [6] Department of Medical Microbiology, Amsterdam Infection and Immunity, Amsterdam University Medical Center, University of Amsterdam, Amsterdam, The Netherlands. [7] Netherlands Reference Laboratory for Bacterial Meningitis, Center of Infection and Immunity Amsterdam, Amsterdam University Medical Center, Amsterdam, The Netherlands. [8] These authors contributed equally: Chrispin Chaguza, Dorota Jamrozy, Merijn W. Bijlsma. [9] These authors jointly supervised this work: Arie van der Ende, Stephen D. Bentley. ✉email: cc19@sanger.ac.uk; a.vanderende@amsterdamumc.nl; sdb@sanger.ac.uk

Group B Streptococcus (GBS), or *Streptococcus agalactiae*, is an emerging β-haemolytic pathogen, which causes substantial neonatal disease burden and mortality worldwide[1]. Global estimates showed that GBS colonises approximately 21 million pregnant women annually, leading to ascending infections associated with approximately 3.5 million preterm births and more than 57,000 fetal infections and stillbirths[2–5]. In neonates and infants, GBS is a cause of approximately 319,000 invasive disease episodes globally on a yearly basis; however, this underestimates the true global disease burden, especially in low-income countries, where little or no data has been reported[3]. These acute invasive neonatal GBS diseases include pneumonia, bacteraemia, and meningitis; broadly classified, based on the time of occurrence, as early-onset disease (EOD) and late-onset disease (LOD), occurring within 0 to 6 and 7 to 89 days after birth, respectively[6–8]. To reduce the risk for vertical transmission of GBS at birth[9–11], risk-based or universal screening and intrapartum antibiotic prophylaxis for pregnant women with risk factors are implemented in the third trimester of pregnancy, particularly in high-income countries[12,13]. Despite this, intrapartum antibiotics are ineffective against GBS-associated stillbirths[14] and LOD, as seen by its increasing incidence globally[7,15,16], and there is conflicting evidence regarding its impact in preventing EOD[7,17,18]. Furthermore, these interventions are less likely to be feasible in low-income settings[19]. Therefore, the World Health Organisation (WHO) has called for developing maternal GBS vaccines[5], widely regarded as the most effective strategy for reducing invasive neonatal GBS diseases[4,14]. However, no vaccine has been licensed to date, although a few candidates are undergoing preclinical development and early-phase clinical trials[20–24].

The sialic acid capsular polysaccharide (*cps*) is the primary virulence determinant for GBS, which promotes immune evasion by inhibiting phagocytosis[25], complement deposition and activation[26], and platelet-mediated killing[26–28]. GBS also contains an arsenal of other virulence factors involved in immune evasion[29–31], toxin-mediated virulence[32,33], transcription regulation[34], and adhesion to the epithelial tissues and host cell entry[35–37]. Except for the *cps* genes, most of the virulence genes are core genes, ubiquitously found across the GBS species. Therefore, the mere presence and absence patterns of these genes are unlikely to explain the inter-strain variability in GBS phenotypes and disease outcomes. However, accessory genes that are variable present, and allelic variation within the core genome, may contribute to inter-strain phenotypic differences and clinical manifestations, such as the onset time of disease and the tissues that are invaded. Although previous studies have reported mutations and lineage-specific genes in GBS[38,39], that potentially affect virulence and niche adaptation, genetic variation in GBS influencing the onset time of acute invasive neonatal disease and meningeal invasion remains poorly understood. Revealing such pathogenicity loci could accelerate the development of diagnostics, therapies, and especially vaccines[1], which are universally considered the most effective strategy to reduce GBS-associated stillbirths, preterm births, and invasive burden and sequelae in neonates globally. The application of agnostic and unbiased comparative genomic analysis approaches, particularly genome-wide association studies (GWAS), has shown remarkable potential for uncovering the genetic basis of bacterial phenotypes[40,41], such as disease susceptibility[42–48], tissue invasion[42,49] and virulence[50], antimicrobial resistance[51–55], and niche adaptation[56–58].

In this study, we performed well-controlled GWAS of an extensive collection of GBS clinical isolates to investigate the genetic basis of the disease onset time and central nervous system (CNS) tissue invasion of GBS isolates in neonates with acute invasive disease. We leveraged a catalogue of 1,338 whole-genome sequences of GBS isolates sampled over thirty years, 1987 to 2016, through a long-term nationwide bacterial surveillance programme in the Netherlands[7]. We show that genomic variation within and outside the capsule biosynthesis locus region influences disease onset time and CNS invasion of GBS in neonates. These findings highlight the critical role of the capsule and other genomic loci in the pathogenicity of GBS, implicating them as potential candidates for the development of capsule- and protein-based vaccines, diagnostics, and treatments to reduce the neonatal GBS burden and mortality globally.

## Results

**Thirty years of invasive neonatal GBS sampling through a national surveillance programme.** We analysed a collection of 1,338 whole-genome sequences of GBS from clinical isolates, sampled from neonates and infants with acute invasive diseases in the Netherlands, to understand the genetic basis for the disease onset time and invasion of the central nervous system (CNS) tissue (Figs. 1a and 2; and Supplementary Data 1 and 2). The isolates were collected over 30 years (1987 to 2016) by the Netherlands Reference Laboratory for Bacterial Meningitis (NRLBM) through well-established national surveillance of meningitis and bacteraemia[59] (Fig. 1b; and Supplementary Fig. 1 and Supplementary Data 1). Of these isolates, 494 were sampled from cerebrospinal fluid (CSF), representing CNS invasion, while 844 were collected from blood or non-CNS site. When the isolates were stratified by disease onset time, 826 isolates were sampled from neonates with EOD, and 515 were associated with LOD (Fig. 1b and Supplementary Fig. 2). We found the ten GBS capsular serotypes known to date[60], 134 sequence types (ST), and six clonal complexes (CC) based on the multilocus sequence typing (MLST) scheme[61], and six previously defined clades or lineages[59,62], based on the genomic sequence clustering algorithms using Bayesian approaches[62] (Fig. 1c and Supplementary Fig. 2). The most common serotype was III, both in EOD (51.76%) and LOD (75.53%), and CNS (78.74%), and non-CNS disease (50.47%). Besides serotype III, serotype Ia was the second most prevalent serotype found in 19.43% of the isolates. The incidence of GBS serotypes and lineages or clades, especially serotype III associated with the clonal complex [CC] 17, increased over time (Fig. 1b–d)[63], consistent with studies regionally[64,65] and globally[66,67]. The increasing trend of this CC17 lineage correlated with the emergence of multidrug resistance[64].

**GBS serotypes and lineages influence disease onset time and meningeal infection.** We first compared the frequency of serotypes, clonal complexes, and lineages among the isolates stratified by the acute neonatal invasive disease onset time and CNS infection. We found five capsular serotypes relatively less common among the isolates associated with LOD than EOD, namely serotype Ia (odds ratio: 0.59, $P = 0.0004$), Ib (odds ratio: 0.45, $P = 0.002$), II (odds ratio: 0.21, $P = 4.47 \times 10^{-07}$), and V (odds ratio: 0.36, $P = 0.0009$) (Fig. 3a, Supplementary Fig. 2, and Supplementary Data 3). In contrast, serotype III showed a higher relative frequency among the LOD than EOD isolates (odds ratio: 2.88, $P = 2.07 \times 10^{-18}$), consistent with the findings elsewhere showing that serotype III is the predominant serotype associated with LOD[65,68,69]. Overall, more LOD isolates were associated with CNS disease compared to EOD (odds ratio: 3.93, $P = 3.39 \times 10^{-31}$), a pattern supported by epidemiological studies[68,70]. Correspondingly, the associations between serotypes and MLST clonal complexes, lineages and clades showed similar patterns (Fig. 3b, c and Supplementary Fig. 2). In terms of the type of the infected tissues, serotype III showed a higher relative

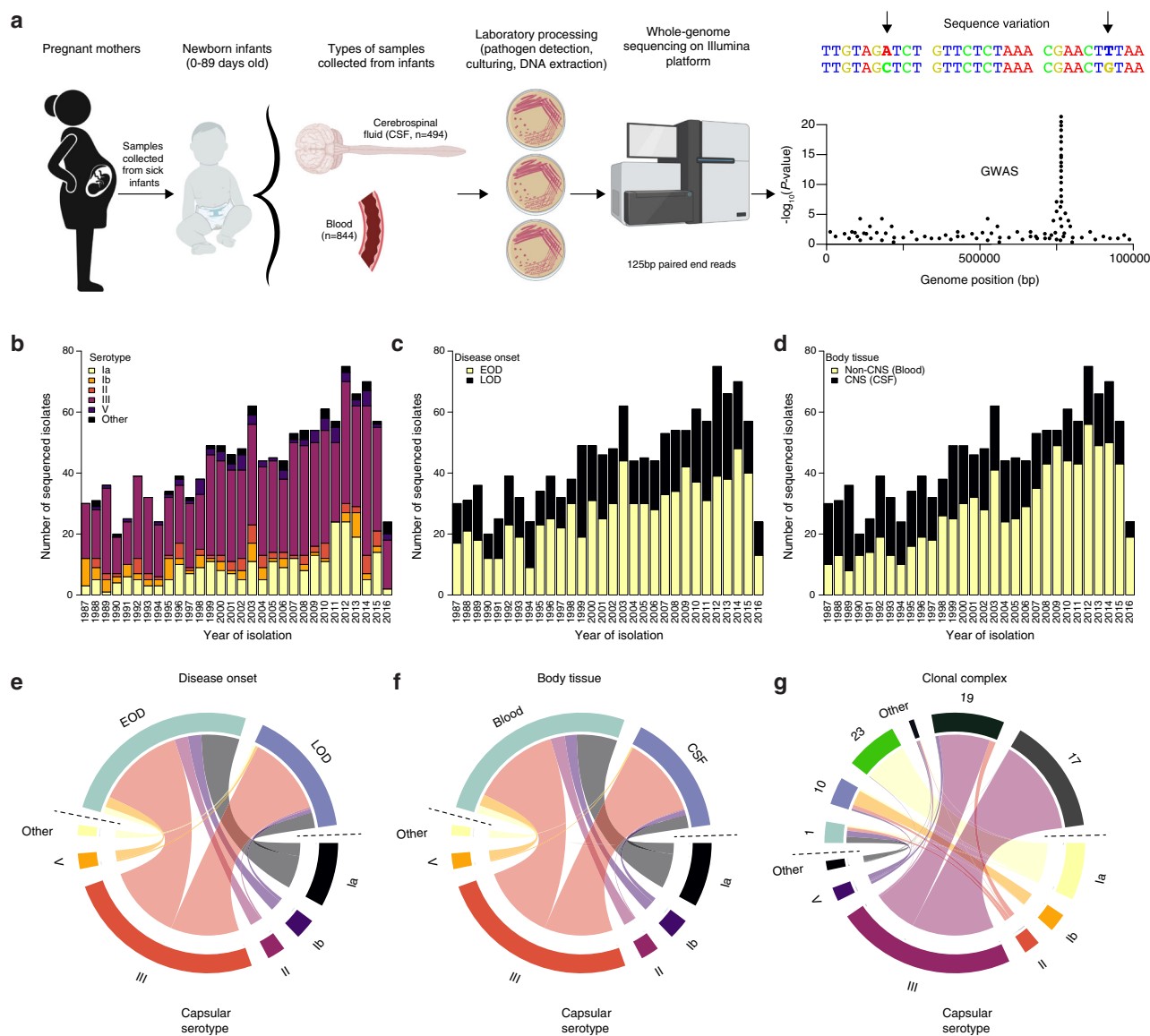

**Fig. 1 Study design and characteristics of the 1,338 GBS isolates sampled from newborns over 30 years. a** Schematic diagram showing sample collection and study design. GBS samples were collected annually for 30 years, 1987–2016, from blood and cerebrospinal fluid (CSF) of newborns aged 0–89 days through a national bacterial surveillance programme in the Netherlands. The samples were processed as described in the methods, and whole genomes were sequenced on the Illumina HiSeq platform with 125 bp reads for downstream analyses, particularly GWAS. The number of sequenced GBS isolates annually is stratified by the **b** proportion of serotypes, **c** proportion of isolates classified as EOD and LOD. **d** proportion of isolates by isolation tissue. The total number of sequenced GBS isolates stratified by **e** disease onset time and capsular serotype, **f** isolation tissue and capsular serotype, and **g** clonal complex and capsular serotype. The icons in panel **a** were created with permission in BioRender.com (https://biorender.com/). Source data used to generate figures in panel **b**–**g** is available in Supplementary Data 2.

frequency among isolates sampled from the CNS than from non-CNS sites (odds ratio: 3.62, $P = 1.75 \times 10^{-25}$). Conversely, the frequency of the other serotypes, except serotype Ib, was lower among CNS isolates than among non-CNS isolates (Fig. 3d–f). These findings showed that the capsular serotype and genetic background of GBS influence the disease onset time and invasion of the CNS.

**GWAS implicates genomic loci influencing GBS disease onset time**. We next investigated whether genomic variation in the GBS genome influences the onset time for acute invasive diseases in neonates and infants (Supplementary Fig. 3). Firstly, we specified the disease onset time as a categorical binary target variable, whereby EOD and LOD were defined as the affected and unaffected status, respectively. To account for the population structure,

which typically confounds bacterial GWAS analyses, if not accounted for[41], we included a pairwise genetic relatedness matrix of the isolates as a random covariate in the linear mixed model. We sequenced the genomes of the isolates using the same protocol and identical read lengths to control for potential batch effects, as unequal read lengths can significantly confound bacterial GWAS[71] (Supplementary Figs. 4 and 5). After correcting for multiple statistical tests, we found no single-nucleotide polymorphism (SNP), in the GBS reference genome (GenBank accession: AP018935.1), statistically associated with the disease onset time (Fig. 4a and Supplementary Data 4). To address the potential impact of frequent genetic exchanges in bacteria via recombination and horizontal gene transfer[72], we next performed the GWAS using the presence and absence patterns of unitigs—unique high-confidence contiguous sequences[73]. The unitigs are advantageous since they

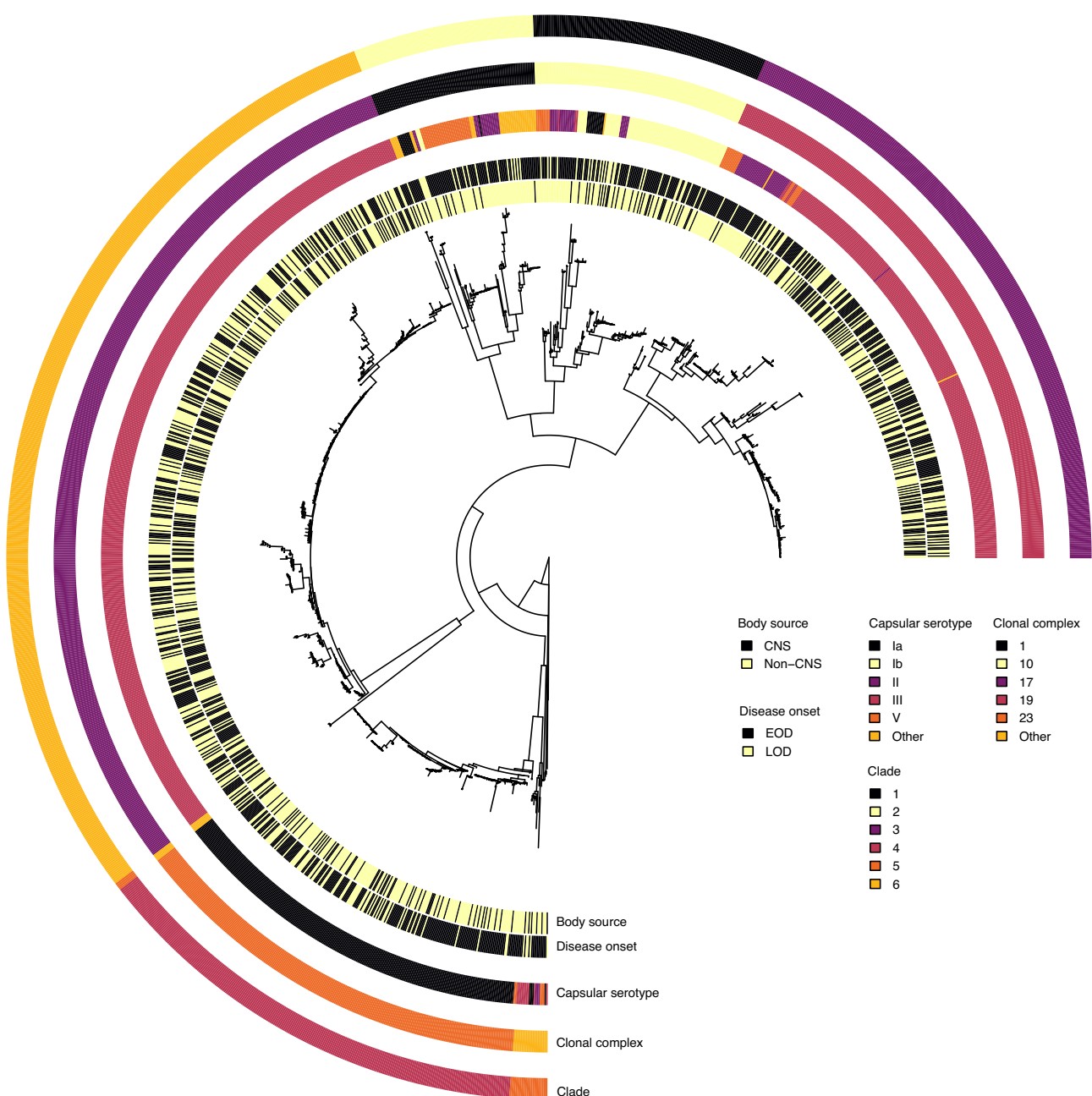

**Fig. 2 Maximum-likelihood phylogenetic tree of 1,338 GBS isolates from the Netherlands.** Each circular ring at the tips of the phylogenetic tree represents the body isolation site or source, disease onset time, capsular serotype, clonal complex based on the MLST approach, and the clade defined based on the Bayesian clustering approach, which yielded concordant groupings with the clonal complex. The phylogeny was rooted using a genome from a related but different species, *Streptococcus pneumoniae*, as an outgroup (not shown in the tree).

efficiently capture inter-strain genomic variation—SNPs, insertions and deletions, and genomic rearrangements—in intergenic and coding regions of the core and accessory sequences[74]. We found a single unitig, located in an intergenic region, statistically associated with the categorical disease onset time of the isolates (odds ratio: 0.76, adjusted $P = 1.38 \times 10^{-07}$) (Fig. 4b and Supplementary Data 4). However, the AP018935.1 reference used for visualisation of the genomic context of the variants did not contain this unitig, therefore, it is not shown in Fig. 4b.

As the categorical classification of the GBS disease onset time is mainly for convenience, clinically, we posited that the GWAS based on the continuous values for the disease onset time would improve the statistical power to identify statistically significant

genotype–phenotype associations (Supplementary Figs. 7 and 8). Therefore, we next repeated the GWAS using the continuous target variable defined as the number of days from birth to disease onset, while similarly controlling the clonal population structure. Since the number of days from birth to disease onset is right skewed, we applied a rank-based inverse normal transformation to generate a normally distributed values to improve the power to uncover associations. We found no unitig sequences with statistically significant association with the transformed disease onset time (Fig. 4c, d). Similarly, we found no association of the accessory genes with disease onset time (Supplementary Fig. 3). The resulting Q–Q plots showed no issues for GWAS analyses due to the population structure (Fig. 4e–h). Therefore, we

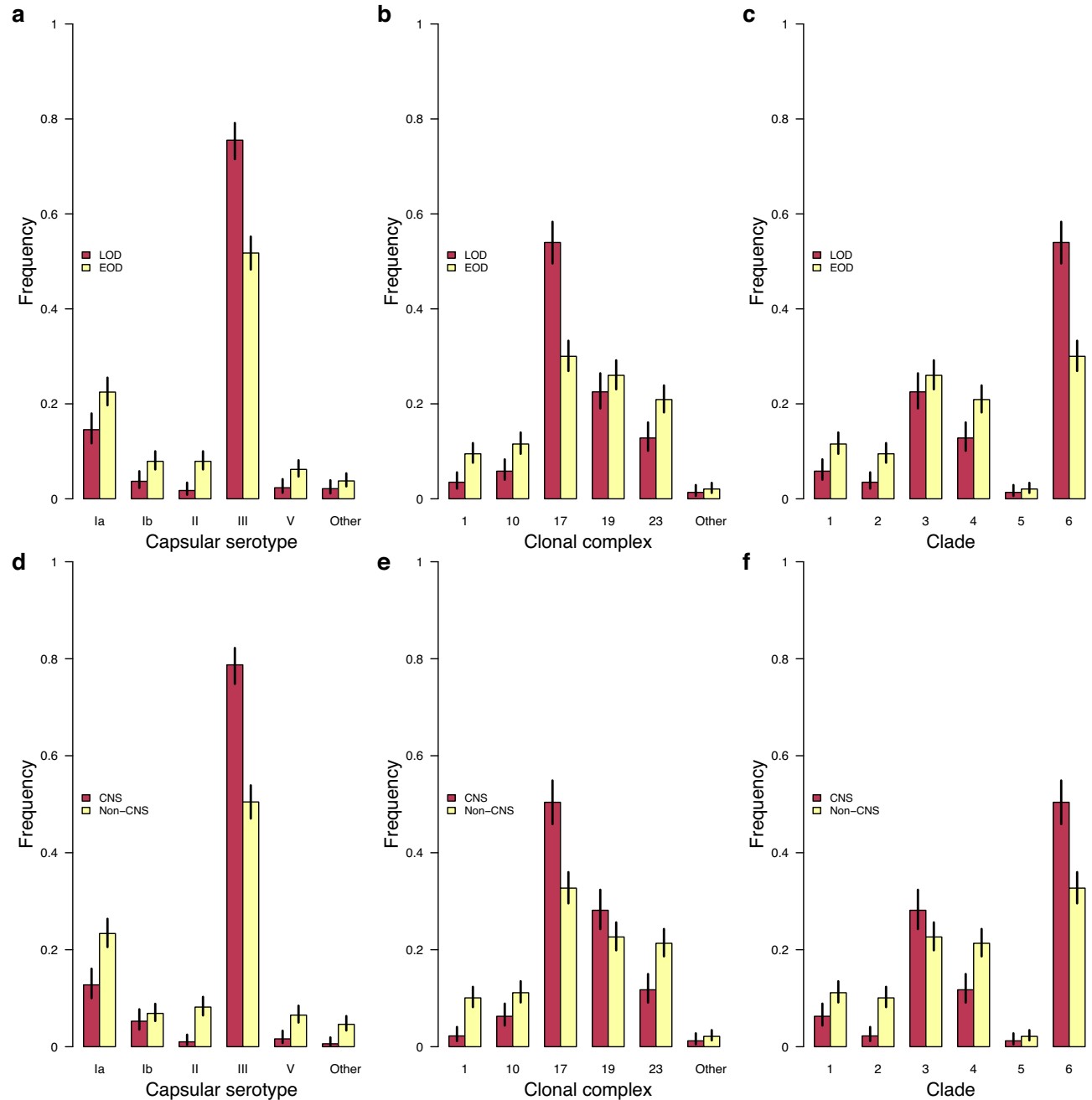

**Fig. 3 Relative frequency of GBS strains stratified by disease onset time and body isolation tissue. a** Relative frequency of GBS capsular serotypes in LOD ($n = 515$) and EOD ($n = 823$) isolates. The statistical significance for each serotype based on the test of given proportions were as follows: Ia ($P = 0.0004$), Ib ($P = 0.0017$), II ($P = 4.47 \times 10^{-07}$), III ($P = 2.07 \times 10^{-18}$), V ($P = 0.0009$), and Other ($P = 0.1082$). **b** Relative frequency of GBS clonal complexes in LOD ($n = 515$) and EOD ($n = 823$) isolates. The statistical significance for each clonal complex based on the test of given proportions were as follows: 1 ($P = 2.75 \times 10^{-05}$), 10 ($P = 0.0005$), 17 ($P = 3.75 \times 10^{-18}$), 19 ($P = 0.1710$), 23 ($P = 0.0002$), and Other ($P = 0.4024$). **c** Relative frequency of GBS clades in LOD ($n = 515$) and EOD ($n = 823$) isolates. The statistical significance for each clade based on the test of given proportions were as follows: 1 ($P = 0.0005$), 2 ($P = 2.75 \times 10^{-05}$), 3 ($P = 0.1710$), 4 ($P = 0.0002$), 5 ($P = 0.4024$), and 6 ($P = 3.75 \times 10^{-18}$). **d** Relative frequency of GBS capsular serotypes in isolates sampled from the CNS ($n = 494$) and non-CNS ($n = 844$) tissues. The statistical significance for each serotype based on the test of given proportions were as follows: Ia ($P = 1.50 \times 10^{-06}$), Ib ($P = 0.2932$), II ($P = 1.32 \times 10^{-09}$), III ($P = 1.75 \times 10^{-25}$), V ($P = 2.23 \times 10^{-05}$), and Other ($P = 1.22 \times 10^{-05}$). **e** Relative frequency of GBS capsular serotypes in isolates sampled from the CNS ($n = 494$) and non-CNS ($n = 844$) tissues. The statistical significance for each serotype based on the test of given proportions were as follows: 1 ($P = 9.87 \times 10^{-09}$), 10 ($P = 0.0034$), 17 ($P = 2.23 \times 10^{-10}$), 19 ($P = 0.0256$), 23 ($P = 7.96 \times 10^{-06}$), and Other ($P = 0.2874$). **f** Relative frequency of GBS lineages in isolates sampled from the CNS ($n = 494$) and non-CNS ($n = 844$) tissues. The statistical significance for each serotype based on the test of given proportions were as follows: 1 ($P = 0.0034$), 2 ($P = 9.87 \times 10^{-09}$), 3 ($P = 0.0256$), 4 ($P = 7.96 \times 10^{-06}$), 5 ($P = 0.2874$), and 6 ($P = 2.23 \times 10^{-10}$). All the error bars in each plot represents 95% confidence intervals. Source data used to generate figures in panel **a–f** is available in Supplementary Data 3.

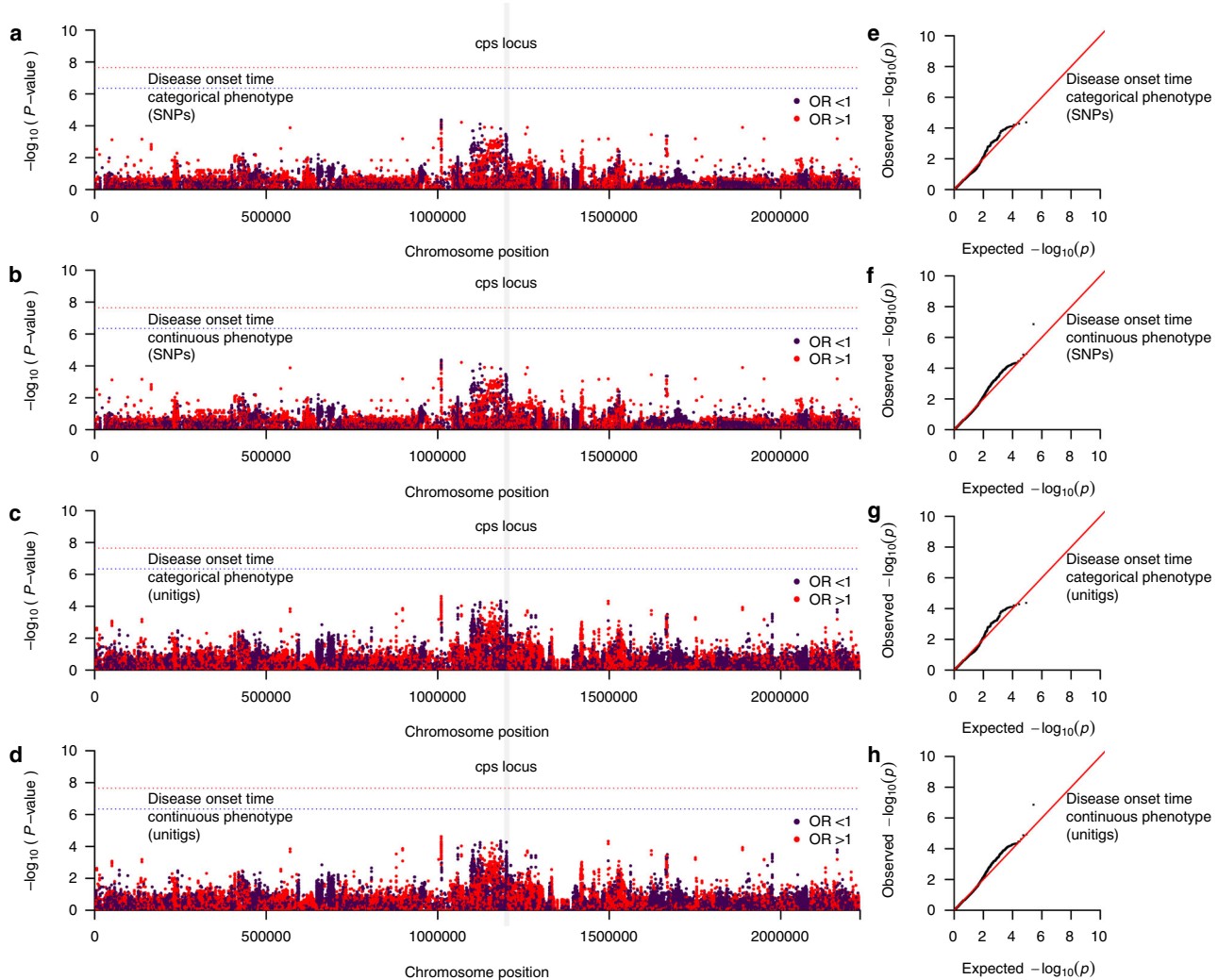

**Fig. 4 Manhattan plots showing the association between GBS genomic variation and GBS disease onset time.** Statistical significance (–log$_{10}$[P-value]) of the GBS genomic variants based on the likelihood ratio test are coloured by the exponentiated fixed effect coefficients or odds ratios of the minor allele as the effect alleles in the GWAS. **a** SNP-based GWAS using disease onset time as a categorical target variable, defined as EOD and LOD. **b** Unitig-based GWAS using disease onset time as a categorical target variable, defined as EOD and LOD. **c** SNP-based GWAS using disease onset time as a continuous target variable defined as the rank-based inverse normal transformation number of days from birth to disease onset. **d** Unitig-based GWAS using disease onset time as a categorical target variable. **e** SNP-based GWAS using the transformed disease onset as a continuous target variable. Q–Q plots showing the relationship between the observed statistical significance and the expected statistical significance, **f** Unitig-based GWAS using disease onset time as a categorical target variable, **g** SNP-based GWAS using the transformed disease onset time as a continuous target variable, and **h** Unitig-based GWAS using disease onset time as a continuous target variable. The red and blue dotted lines represent the genome-wide significance and suggestive threshold, respectively. The variants with odds ratios (OR) > 1 is coloured in red while those with odds ratio <1 is coloured in dark purple. Source data for panel **a–f** is available in Supplementary Data 4 and on GitHub (https://github.com/ChrispinChaguza/GBS_Study_NL).

concluded that specific genomic loci in the GBS genomes had minimal influence on the disease onset time of acute invasive neonatal diseases.

**Genetic variation within the capsule locus influences meningeal invasion of GBS.** Considering the differences in the relative frequency of capsular serotypes in the isolates sampled from the CNS and non-CNS sites (Fig. 3), we next investigated differences in the abundance of genetic variation in the isolates sampled from the CNS and non-CNS tissues, which could influence meningeal tissue invasion of GBS. Similarly, we performed a GWAS with sampled tissue as the target variable, coding samples isolated from the CNS as the affected status and from non-CNS sites as the unaffected status. As done in the GWAS for the disease onset time (Fig. 4), we controlled the population structure to account

for the clonality of the isolates. We identified four SNPs from the GWAS based on SNP genetic variation (Fig. 5a and Supplementary Data 4). These SNPs were in genes within the *cps* locus, with locus tags SAGS_1212 (*epsJ_2*) (odds ratio: 0.86, adjusted $P = 5.14 \times 10^{-20}$) and three in SAGS_1213 (odds ratio: 0.85, adjusted $P = 0.045$) in the GBS genome. Interestingly, we implicated nine *cps* genes in the GWAS using the genetic variation captured by the unitig sequences, including genes identified in the SNP-based GWAS (Fig. 5b, Table 1, and Supplementary Data 4). These unitigs were in several genes, all within the *cps* locus region (odds ratio: 0.85 to 0.86, adjusted $P = 2.30 \times 10^{-20}$ to 0.04). These genes included locus tags SAGS_1212 (*epsJ_2*) and SAGS_1213. We also identified 77 suggestive hits in several genes within and outside the *cps* locus, including SAGS_1212 (*epsJ_2*), SAGS_1213, SAGS_1226 (*arsC*), SAGS_1214, SAGS_1209, SAGS_1200 (*parC*), SAGS_1201 (*parE*), SAGS_1207 (*neuC*) and SAGS_1228 (*rpiA*).

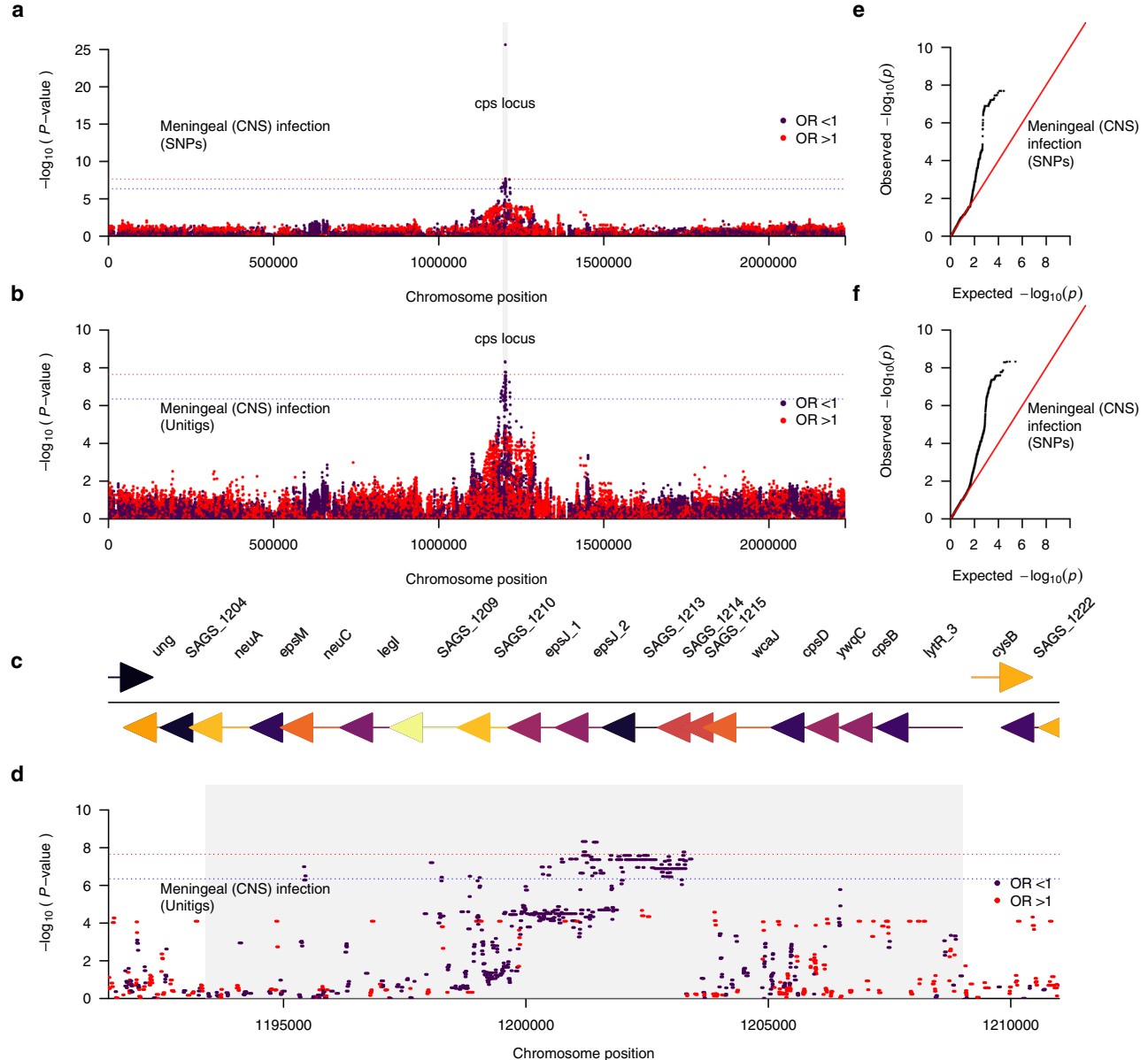

**Fig. 5 Manhattan plots showing the association between GBS genomic variation and the CNS infection status.** Statistical significance (−log₁₀[P-value]) of the GBS genomic variants based on the likelihood ratio test are coloured by the exponentiated linear mixed model coefficients or odds ratio as the target variable in the GWAS. **a** SNP-based GWAS and **b** unitig-based GWAS using CNS infection status of the GBS isolates as the target variable specified as isolation from the CNS or non-CNS tissue. **c** Genomic features in the GBS capsular polysaccharide (cps) locus, and **d** corresponding Zoom plot showing the statistical significance and location of the genetic variants (unitigs), which mapped to the cps region of the complete GBS reference genome (GenBank accession: AP018935.1). Q–Q plots showing the relationship between the observed statistical significance and the expected statistical significance for **e** SNP-based GWAS, and **f** Unitig-based GWAS using body isolation tissue as the target variable. The red and blue dotted lines represent the genome-wide significance and suggestive threshold, respectively. The variants with odds ratios (OR) > 1 is coloured in red while those with odds ratio <1 is coloured in dark purple. Source data for panel **a**–**f** is available in Supplementary Data 4 and on GitHub (https://github.com/ChrispinChaguza/GBS_Study_NL).

Additionally, we identified a total of 123 suggestive unitigs mostly mapping to the cps locus and other genomic region (Supplementary Data 4). The cps genes tagged by the unitigs included those encoding a multidrug major facilitator superfamily (MFS) transporter (cpsG), glycosyltransferase CpsJ (cpsJ), a capsular polysaccharide biosynthesis protein CpsA (cpsA), and a polysialic acid biosynthesis protein P7 (neuC) while the other genes flanking the cps locus, included DNA topoisomerase 4 subunit A (parC), DNA topoisomerase IV subunit B (gyrB) and an arsenate reductase (arsC) (Fig. 5c). Consistent with the GWAS of the disease onset time outcome, the Q–Q plots for the analysis based on the CNS infection phenotype suggested no apparent issues

when controlling the population structure (Fig. 4e, f). Complementary GWAS based on the presence and absence of accessory genes revealed a single statistically significant association for a phosphopentomutase encoding gene, and two suggestive hits for capsular biosynthesis genes (Supplementary Fig. 6c). Altogether, these findings suggested that genetic variation within the capsule biosynthesis locus influences bloodstream-to-meningeal tissue invasion and potentially survival in the CNS.

**Genetic variation for CNS invasion varies by GBS serotype.** To understand the distribution of identified genetic variants within

**Table 1 Summary of the unitig sequences statistically associated with infection of the CNS in the GWAS using FaST-LMM.**

| Locus tag | Gene name | Reference genome | Number of unitigs | P-value range | Adjusted P-value (Q) range | Odds ratio range | Gene product |
|---|---|---|---|---|---|---|---|
| CHF17_01256 | cpsG | CP022537.1 | 1 | $1.696 \times 10^{-08}$ | 0.038 | 0.845 | Glycosyl transferase CpsG(V) |
| No match | No match | No match | 1 | $1.361 \times 10^{-08}$ | 0.030 | 0.850 | No match |
| AV644_06110 | | CP013908.1 | 1 | $4.745 \times 10^{-09}$ | 0.011 | 0.857 | Arsenate reductase |
| BB165_05995 | | CP021870.1 | 1 | $5.086 \times 10^{-09}$ | 0.011 | 0.866 | Capsular biosynthesis protein |
| CCZ24_04050 | | CP021773.1 | 1 | $1.656 \times 10^{-08}$ | 0.037 | 0.847 | Capsular biosynthesis protein |
| CWQ20_06175 | | CP025029.1 | 3 | $4.706 \times 10^{-09} - 5.086 \times 10^{-09}$ | 0.011-0.011 | 0.864-0.866 | Capsular biosynthesis protein |
| GBS222_1012 | | FO393392.1 | 1 | $1.617 \times 10^{-08}$ | 0.036 | 0.843 | Hypothetical protein |

the *cps* locus associated with CNS invasion, we compared the relative abundance of the *cps*-associated unitigs in the GBS isolates sampled from the CNS and non-CNS tissues stratified by serotype. We found certain unitigs were differentially abundant among the isolates sampled from the CNS and non-CNS tissues, especially for serotype IV and non-typeable isolates;[75,76] the latter account for ~10% of the GBS isolates in Europe[77] (Supplementary Fig. 9). These two serotypes were relatively less abundant in the GBS isolates in the Netherlands. However, the rest of the unitigs showed similar abundance among the CNS and non-CNS isolates for the other serotypes. Therefore, these findings suggested that the genetic variants had a small effect on invading the CNS tissue.

**Heritability highlights a moderate effect of genetics on disease onset time and CNS invasion.** We then formally quantified the variability in the neonatal disease onset time and CNS infection phenotypes explained by the genetic variability in the GBS genomes. To achieve this, we estimated the narrow-sense heritability ($h^2$) for each phenotype using several methods, namely GEMMA[78], FaST-LMM[79], and GCTA[80] (see methods). Previous GWAS of bacteria suggested a negligible contribution of pathogen genetics to the variability in the phenotypes associated with disease outcomes, for example, severity[44,81]. Contrary to colonisation, genetic variation correlated with invasive disease is unlikely to be positively selected by natural selection, because invasive disease is an evolutionary dead-end. Essentially, either the host immune system clears the pathogen, or the host dies without impacting onward transmission of the pathogen and frequency of the variants in the population[82], which obscures the genetic–phenotype association signal in the GWAS. Therefore, we hypothesised that the disease onset time and CNS infection phenotypes have low heritability. Consistent with our hypothesis, we found low estimates for the narrow-sense heritability using GEMMA for the disease onset time as categorical ($h^2 = 0.07$ to 0.21) and continuous ($h^2 = 0.06$ to 0.21) variables, and CNS invasion ($h^2 = 0.06$ to 0.14) based on different types of genetic variation (Fig. 6a and Supplementary Data 5). In support of these findings, we found similar heritability estimates using FaST-LMM[79] (Fig. 6b and Supplementary Data 5), although slightly lower values were inferred with GCTA[80] (Fig. 6c and Supplementary Data 5). Overall, these findings suggested a modest but non-negligible impact of GBS genetics on the inter-strain variability in the disease onset time and infection of the CNS tissue.

**Discussion**
This study leveraged an extensive collection of acute invasive neonatal GBS clinical isolates routinely collected over thirty years through a robust and well-established national bacterial surveillance programme in the Netherlands[83]. By applying well-controlled linear mixed model GWAS approaches, we have systematically identified genomic variation in GBS associated with the disease

onset time and invasion of the CNS. These findings suggest that pathogen genetics modulates the timing of GBS disease in neonates and bloodstream-to-meningeal invasion, which increases the risk for meningitis—a severe clinical manifestation of GBS disease typically associated with long-term neurologic sequelae[84] and mortality[63]. Previous studies have not implicated the loci associated with disease onset time identified in this study with GBS pathogenicity and virulence, highlighting the utility of agnostic and unbiased GWAS approaches to unravel novel genotype–phenotype associations. Furthermore, although the GBS capsule is a well-known virulence determinant critical for immune evasion and virulence[25–28], our results provide the evidence that genomic variation within the capsule biosynthesis locus influences the pathogenesis of meningitis by modulating bloodstream-to-meningeal invasion of the CNS compartment.

Bloodstream infection is the predominant transient invasive disease state preceding infection of the CNS, especially meningitis. Therefore, such an aetiology of meningitis implies that GBS isolated from the CNS are also capable of causing bloodstream infection. However, the converse may not necessarily hold if the pathogen genetics influenced CNS invasion. Our findings show that certain GBS isolates infecting the CNS harbour genetic variation within the capsule biosynthetic locus, which may influence meningeal invasion by modulating translocation across the blood-brain-barrier into CNS, possibly through interactions with the host endothelial cells, as similarly seen with other adhesins and host transmembrane receptors[85]. Similar to other encapsulated bacteria, such as *Streptococcus pneumoniae*[86], the polysaccharide capsule of GBS is a critical virulence determinant[87], which promotes immune evasion by inhibiting phagocytosis[25], complement deposition, and activation[26], and platelet-mediated killing[26–28]. The *cps* genes containing genetic variation associated with CNS infection included those encoding for a transcriptional regulator for the *cps* operon (*cpsA*), synthesis and transport of oligosaccharides to the outside of the cell membrane (*cpsJ* and *cpsG*), and synthesis transport of sialic acids (*neuC*)[88]. Therefore, the identified genomic variation associated with the CNS invasion may also promote GBS survival in the CNS, ultimately modulating the risk of meningitis. However, we found no statistically significant genetic variation tagging other known virulence factors important for meningeal tropism, such as HvgA[89], which suggest although such genes are generally essential for GBS meningeal tropism, their allelic variability does not influence the ability of GBS to invade the CNS. Our findings highlight the importance of the *cps* and other genomic loci in the pathogenicity and virulence of GBS, implicating them as potential targets for the development of capsule- and protein-based vaccines, treatments, and diagnostics to reduce the short- and long-term neonatal GBS disease burden and death toll globally. Such GBS vaccines are currently undergoing preclinical[20] and early phase I and II clinical trials[20–23]. Reassuringly, a capsule-based vaccine, for example, a hexavalent polysaccharide-protein conjugate vaccine

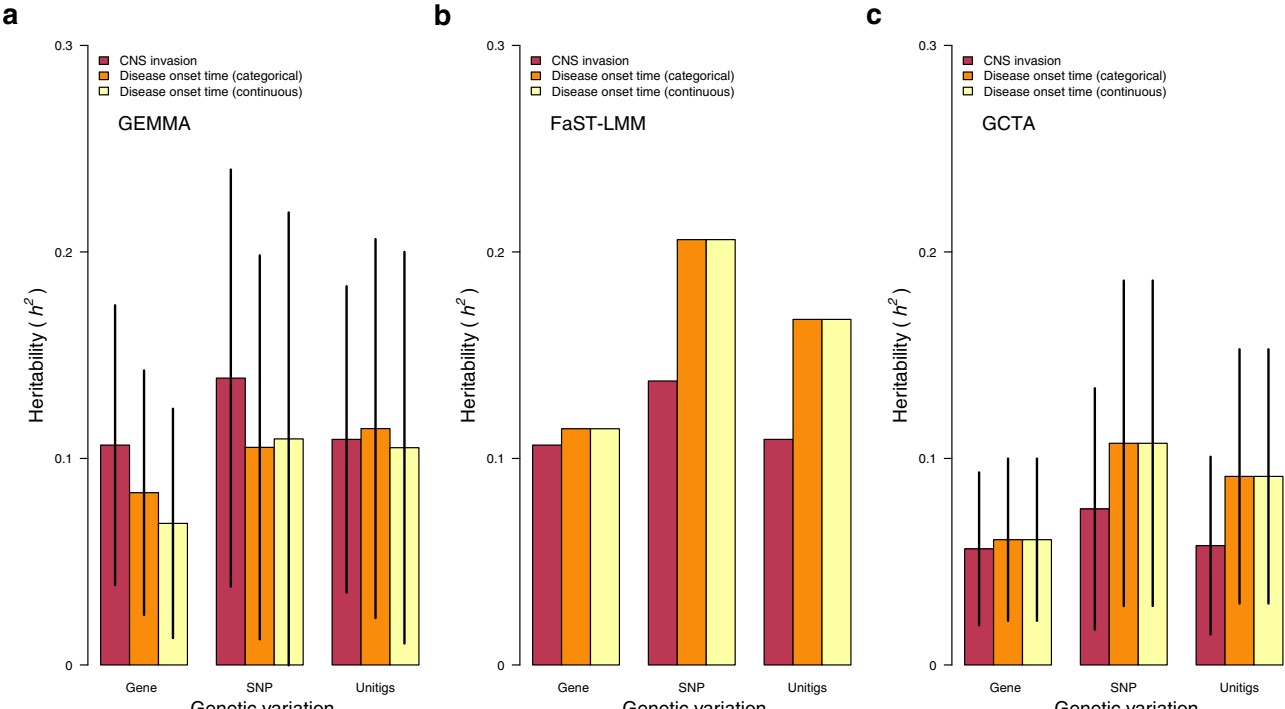

**Fig. 6 Estimates of the narrow-sense heritability ($h^2$) using different genetic variants and methods. a** Heritability estimated for accessory genes, SNPs, and unitigs using GEMMA. **b** Heritability estimated for accessory genes, SNPs, and unitigs using FaST-LMM. **c** Heritability estimated for accessory genes, SNPs, and unitigs using GCTA. FaST-LMM reported no standard errors for the heritability estimates, therefore, we do not show the confidence intervals in the panel **b**. Heritability is expressed as a proportion with values ranging from 0 to 1. The error bars in each plot represents 95% confidence intervals. The estimates in panels **a–c** are based on $n = 1338$ isolates. Source data used to generate figures in panel **a–c** is available in Supplementary Data 5.

formulation targeting GBS serotypes Ia, Ib, II, III, IV, and V, could target between 93 to 99% of isolates from maternal colonisation, maternal invasive disease, and neonatal and infant invasive disease;[60] potentially reducing preterm and stillbirths, neurologic sequelae, mortality, and economic burden globally[84].

Although GBS genetics appears to substantially affect the disease onset time and CNS invasion, the amount of variability in these phenotypes accounted for by the genomic variation, i.e., the narrow-sense heritability, appears to be moderate but not negligible. Such heritability estimates reflect a polygenic nature of the phenotypes, as complex traits are modulated by multiple variants with potentially small effect sizes, requiring larger datasets to implicate them in the GWAS and account for the missing heritability. These findings contrast with the genomic variation associated with other bacterial phenotypes, including antimicrobial resistance[90], host niche adaptation[57], and invasiveness[44,46], which typically exhibit high heritability reflecting substantial natural selection pressures on these genomic loci. However, a current challenge remain that an unknown proportion of bloodstream infection cases result in an unidentified meningitis, i.e., some of the cases with a positive blood culture would also have had a positive CSF culture if a lumbar puncture would have been performed, which is not done in all neonates with signs of infection. Therefore, the GBS isolates sampled from the CNS and non-CNS sites may not be completely genetically distinguishable likely dampening the genotype–phenotype association signal and heritability. Altogether, these findings suggest that although pathogen genetics partly influence the variability in the onset of acute invasive GBS disease and tissue invasion in the neonates and infants, the maternal and neonatal host factors, host-pathogen interactions, and the environmental factors contribute to a more considerable extent to the GBS disease pathogenesis.

This study shows that genomic variation of GBS, within and outside the capsule biosynthesis locus, influences the disease onset time and bloodstream-to-meningeal translocation and invasion of the CNS in neonates with acute invasive disease. These findings highlight the crucial role of the sialic acid polysaccharide capsule in the virulence of GBS, emphasising the need for evaluating capsule-based vaccines to prevent and control invasive diseases in neonates and infants. Furthermore, our study highlights the utility of microbial population genomics combined with well-established clinical bacterial surveillance programmes to generate novel and unbiased insights into the contribution of bacterial genetics to the population-level pathogen traits, potentially challenging to study experimentally, to inform disease prevention and control strategies. As more GBS whole-genome sequences, as well as strain and patient-level meta-data, become available, we are optimistic that exploiting these big data by applying robust GWAS and possibly machine learning approaches, as highlighted in this study, will not only validate but also unravel additional novel cryptic pathogenicity loci influencing several GBS phenotypes and clinical outcomes, including mother-to-child transmission, human intestinal and vaginal niche, animal adaptation, and disease severity.

Though this work provides robust evidence of the contribution of GBS genetics to disease onset and CNS invasion, there are some limitations worth noting. Meningitis diagnosis could have been missed in some sepsis cases because a lumbar puncture was postponed, not done, or false negative, which could result in an underestimation of meningitis incidence. Therefore, although the CSF-positive cases were from meningitis cases, some blood culture isolates are likely from patients with unidentified meningitis, which may have dampened the associations in the GWAS of CNS and non-CNS isolates. A nationwide guideline recommending intrapartum antibiotic prophylaxis to women with risk factors of a newborn with early-onset disease was implemented around 1998; however, this is unlikely to introduce bias in the dataset. We have previously used the concurrent *Escherichia coli* collection to

make the case that there had been no apparent shift in laboratory surveillance practice[7]. We have also compared meningococcal submissions to the reference laboratory to another mandatory notification system and found a similar pattern over time[83]. Furthermore, detailed clinical background information was not available, which restricted the adjustments in the GWAS analyses.

In conclusion, we have shown using population genomics of an extensive and well-sampled collection of neonatal GBS clinical isolates that variation in the GBS genomes, within and outside the capsule biosynthesis region, influence the onset time for acute invasive disease and invasion of the meningeal tissue, highlighting a genetic basis for the inter-strain variability of the GBS disease outcomes in neonates and infants.

## Methods

**Samples, microbiological processing, and ethical approvals.** All GBS isolates cultured from cerebrospinal fluid or blood from patients were submitted to the National Reference Laboratory of Bacterial Meningitis (NRLBM) at the Amsterdam UMC, University of Amsterdam, for further typing and storage, as part of the continuous surveillance of bacterial meningitis in the Netherlands. One thousand and three hundred thirty-eight GBS isolates were selected from a dataset of isolates collected from nationwide surveillance of infants with bacterial meningitis and bacteraemia conducted by the NRLBM[59] (Supplementary Data 1). The isolates were collected over 30 years from January 1987 to January 2016. Of these isolates, 823 and 515 were from neonates with EOD (0 to 6 days, post-birth) and LOD (6 to 89 days, post-birth). The age of the infant was estimated as the time from birth to sample collection as previously described[59]. By isolation source, 494 isolates were sampled from the cerebrospinal fluid (CSF), while 844 isolates were from non-CSF sites, namely blood ($n = 844$). For the present study patient data were anonymized. Additional institutional review board approval is not required for studying submitted strains with anonymised patient data.

**Whole-genome sequencing and molecular typing of GBS isolates.** Genomic DNA was extracted using the Wizard® Genomic DNA Purification Kit from Promega following the manufacturer's instructions[59]. The genomic libraries were created using the Illumina protocol, and whole genomes were sequenced with 125 bp reads on the HiSeq 2000 platform (Illumina, CA, USA). Genome assembly was done using SPAdes genome assembler (version 3.14.0)[91]. The serotype for each isolate was determined in silico using whole-genome sequence data[92,93]. Sequence typing using the multilocus sequence typing (MLST) scheme for GBS[61] was done based on the sequencing data using SRST2 (version 0.2.0)[94].

**Phylogenetic and population structure analysis.** A multi-sequence whole-genome alignment was generated based on consensus sequences of each isolate inferred after mapping reads against a complete GBS reference genome for an invasive human strain HU-GS5823 (GenBank accession: AP018935.1) belonging to sequence type (ST335) and serotype III using Snippy (version 4.6.0) (https://github.com/tseemann/snippy). The genomic positions in the consensus sequences containing variable nucleotide sites or SNPs were extracted from the alignment as multi-FASTA, and variant call format (VCF) files using SNP-sites (version 2.3.2)[95]. The identified SNPs were then used for population structure analysis to identify sequence clusters or lineages using the hierarchical clustering approach implemented in BAPS (version 6)[96]. A maximum-likelihood phylogenetic tree of the entire GBS isolates was generated based on the whole-genome SNP alignment using the general time-reversible (GTR) and Gamma model in FastTree (version 2.1.10)[97,98]. We used a *Streptococcus pneumoniae* strain (ENA accession: ERS812015) as an outgroup to root the inferred phylogenetic tree of the GBS isolates. Visual exploration and analysis of the phylogenetic trees was done using the APE package (version 4.3)[99]. Annotation of the phylogenetic tree with the isolate metadata was done using the "gridplot" and "phylo4d" functions in phylosignal (version 1.3) and phylobase (version 0.8.6) (https://cran.r-project.org/package=phylobase) packages, respectively[100].

**Generating variant data for bacterial GWAS.** The input data for the GWAS were generated using only bi-allelic SNPs in the VCF file of each GBS isolate using VCFtools (version 0.1.16)[101]. The SNPs with minor allele frequency <5% or missingness >5% were filtered out from the final dataset to exclude rare variants using PLINK (version 1.90b4)[102]. To generate the input dataset for the GWAS using the presence and absence patterns of the accessory genes, we first clustered the predicted gene sequences predicted using Prokka (version 1.11)[103] into clusters of orthologous genes (COGs) with Panaroo (version 1.2.2)[104]. We specified the moderate stringency mode when running Panaroo. The COGs are referred to as genes for simplicity. The presence and absence patterns of the predicted genes were merged with the isolate metadata and converted to the pedigree format for the GWAS. Similar to the SNP variant data, the genes with minor allele frequency <5% were filtered out using PLINK (version 1.90b4)[102]. To identify the maximal unitig sequences, i.e., non-branching paths in a compacted De Bruijn graph, we first build

the graph for the entire dataset based on 31 bp k-mer sequences using Bifrost (version 1.0.1)[73]. The unitig sequences generated based on the entire isolate collection were queried against a De Bruijn graph of each genome using Bifrost to determine the presence and absence patterns of each unitig sequence in the genomes. A unitig was considered present when exact matches for all the k-mers in the query unitig sequence were found in each isolate's genome graph. The presence and absence patterns of the unitigs were merged with the affection status (disease onset time and CNS infection) to generate the pedigree data files required for the GWAS. Similarly, the unitigs with minor allele frequency <5% were filtered out using PLINK before the GWAS. The genes and unitigs were not filtered based on missingness as missingness for any reason was regarded as the absence of the gene since it was not possible to distinguish missingness due to either sequencing or assembly errors from true absence due to the variability in the accessory genome.

**Genome-wide association analysis.** We first compared the relative frequency of capsular serotypes, clonal complexes, and lineages in the GBS isolates associated with EOD and LOD and CNS and non-CNS tissue, using Fisher's exact test. We used the clonal complexes and lineages previously defined by Jamrozy et al.[59]. To identify genomic variation, defined in terms of the presence and absence patterns of SNPs, unitig and accessory gene sequence, associated with the GBS disease phenotypes, namely disease onset time, either as a categorical (EOD and LOD) or continuous (days from birth to disease onset) values, and infection of the CNS (CNS and non-CNS) where GBS was isolated; we performed univariate GWAS using robust linear mixed models implemented in FaST-LMM (FastLmmC, version 2.07.20140723)[79]. We applied applied a rank-based inverse normal transformation to the disease onset time as a continuous phenotype to generate normally distributed values as required by the GWAS methods (https://github.com/ChrispinChaguza/GBS_Study_NL). We specified a genetic relatedness matrix of the GBS isolates generated using the unitigs as a random covariate to account for the clonal population structure during the GWAS analysis. Since bacterial chromosomes are haploid, we coded the genotypes as originating from the human mitochondrial genome, designated as chromosome 26, as previously described[45,52]. We adjusted the raw P-values for each variant, inferred using the likelihood ratio test using the Bonferroni correction method to control the false discovery rate due to multiple testing. Since the frequency of genomic variants tested, i.e., accessory genes, SNPs, and unitigs, varied greatly, we used a fixed value for the GBS genome size to represent the possible maximum number of realised genomic variants. This approach is more conservative than adjusting based on the observed variants, minimising false positives but potentially increasing false negatives slightly. However, crucially, our approach ensures the use of a consistent P-value threshold when assessing the statistical significance of different types of genomic variation.

Genetic variants with P-values < $2.24 \times 10^{-08}$, i.e., $\alpha/G$ where the statistical significance threshold $\alpha = 0.05$ and the genome size $G = 2,231,314$ bp for the GBS reference genome of the strain HU-GS5823, were deemed statistically significant. Similarly, variants with P-values < $8.12 \times 10^{-07}$, i.e., $\alpha/G$ where the statistical significance threshold $\alpha = 1$, were considered suggestive. The statistical significance was assessed to compare the expected and observed P-values to visually check potential issues with the population structure, using the quantile-quantile plots generated with qqman (version 0.1.7)[105]. The overall proportion of phenotypic variability explained by variation in the genome (narrow-sense heritability) was estimated using FaST-LMM, GEMMA (version 0.98.1)[78], and GCTA (version 1.93.2)[80]. The genomic features associated with each SNP, accessory gene, and unitigs were identified by comparing them with a panel of GBS reference genomes using BioPython (version 1.78)[106]. In addition, we used BLASTN (version 2.5.0+)[107] to identify genomic regions containing the gene and unitig sequences. The GWAS results were summarised to visually identify genomic regions containing statistically significant genotype–phenotype associations using Manhattan plots generated in R (version 4.0.3) (https://www.R-project.org/).

**Reporting summary.** Further information on research design is available in the Nature Research Reporting Summary linked to this article.

## Data availability

The sequence reads for the isolates used in this study are available in the European Nucleotide Archive under study accession code PRJEB14124. The accession numbers and information for individual isolates are provided in Supplementary Data 1. The authors declare that all other data supporting the findings of this study are available within the paper and its supplementary information files. Additional data for the SNPs, accessory genes, and unitig sequences used in this study are available at https://github.com/ChrispinChaguza/GBS_Study_NL.

## Code availability

All tools and methods used for the analysis are publicly available and fully described in the Methods section. The scripts used in this analysis are available at https://github.com/ChrispinChaguza/GBS_Study_NL.

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

## Acknowledgements

The authors would like to thank the study participants and guardians, the clinical and laboratory staff who collected and processed the samples at various laboratories in the Netherlands, and the sequencing, core, and pathogen teams, and the Bentley lab at the Wellcome Sanger Institute for their support and feedback on the genomic analysis. We would also like to thank Dr. John Lees at European Bioinformatics Institute for providing advice on the GWAS. The study was funded by the Meningitis Research Foundation Project grant 1502.0 (A.v.d.E.), Wellcome Trust grant 098051 (D.J., S.D.B.), Netherlands Organization for Health Research and Development (ZonMw; NWO-Vici 918.19.627 (D.v.d.B.), Amsterdam Medical Centre Innovation grant (D.v.d.B.), and the Bill and Melinda Gates Foundation grant for the Juno project (SDB) [https://www.gbsgen.net/]. C.C. and S.D.B. were supported by funding from the Joint Initiative for Antimicrobial Resistance (JPIAMR) grant no. MR/R003076/1 (S.D.B.), the Bill and Melinda Gates Foundation grant number OPP1034556 (S.D.B.), and Wellcome Trust (2016–2021 core award grant no. 206194).

## Author contributions

C.C., D.J., M.W.B., and S.D.B. conceived the study. A.v.d.E., T.W.K., M.W.B., and D.v.d.B. collected and processed the samples for whole-genome sequencing. D.J. and S.D.B. oversaw whole-genome sequencing. C.C. and D.J. conducted the analysis. D.J., M.W.B., and S.D.B. contributed to the data interpretation and discussions. D.J., A.v.d.E., D.v.d.B., T.W.K., and S.D.B. administered the project. D.J., S.D.B., D.v.d.B., A.v.d.E., and S.D.B. acquired funding for the study. C.C., D.J., and S.D.B. drafted the first version of the manuscript. The manuscript was reviewed by C.C., D.J., A.v.d.E., T.W.K., M.W.B., D.v.d.B., and S.D.B.

## Competing interests

The authors declare no competing interests.
