## [Peer Review File · Nature Communications]

Reviewers' Comments:

Reviewer #1:

Remarks to the Author:

In this manuscript, Chaguza et al. use statistical genetics to test if neonatal infections caused by Group B Streptococcus (GBS) are modulated by bacterial genetic variants. They use a large longitudinal cohort of >1'300 GBS strains collected over 30 years in the Netherlands, for which the date of onset of the disease was recorded, as well as the isolation source. Using various classes of genetic variants and association methods, they find a few variants associated with either disease onset time and with isolation from cerebrospinal fluid.

I found the analysis to be appropriate and the results to be of interest. The heritability estimates for the target phenotypes (both related to infection severity) seem to be comparable with other recent studies. I have some comments regarding the data and the methods applied, and a few major annoyances with the presentation of the data.

Major comments

Are there any known biases in this dataset? As strains and data has been collected for over 30 years, there might have been changes in protocols and similar changes. If that was the case it might be worth it to add a covariate to the analysis.

Related to this, the authors report that the number of meningeal infections may be underestimated due to lack of lumbar sampling. Is there any way that this bias could be estimated by the authors and how this would affect their resulsits?

If I understood correctly, the population structure correction was performed using a kinship matrix derived from the core genome phylogeny. Has it been corrected for the presence of recombination? ALternatively, a covariance matrix from the unitigs presence/absence matrix could be used, as that should be able to account for core and accessory genome population structure.

Have the authors tried to run a burden test to see if any rare variants are associated with infection severity? Their sample size could be sufficient to see some associations.

Can the association with a linear mixed model be run with a continuous phenotype that is not normally distributed? If not, the authors could look into approaches such as warped LMM (doi: 10.1038/ncomms5890). Judging by Supp. Fig. 7 the onset time is indeed not normally distributed.

While the data and methodology are overall sound, the manuscript has several sloppy errors, which one would expect would have been caught by at least one of the 7 authors of this manuscript in the last round of editing before submission. They made reading this manuscript very frustrating. These would necessarily need to be corrected, and I would suggest the authors to double check everything to make sure none of the results, figures, and tables contain any other error. Here's a list of the errors I could spot:

- Line 105: Fig. 2b does not exist (that figure has a single panel), so I'd imagine the authors are referring to Supp. Fig. 2b?
- Line 117, the figure number is missing
- Line 126: Figure 3 is referenced before Figure 2, which is actually never referenced in the whole manuscript
- Supplementary Figure 2d does not make much sense, as the color and columns are the same variable?
- The caption for Figure 4b is incorrect, as the phenotype should be binary here, and not continuous
- Supplementary Figure 6 contains "volcano" and not "violin" plots
- Line 177: this sentence lacks a verb
- Figure 5: the caption contains panels a to f but in the actual figure one panel identifier is missing (probably c is skipped). Also, what do the colors in this panel mean?
- I could not understand what is the difference between Supp. Fig. 4 and Supp. Fig. 5? The plots

look slightly different, but the caption is the same

- Supp. Fig. 8 lacks a legend for the colors
- Supp. Fig. 10: same as Supp. Fig. 6
- Supp. Fig. 11: same as Fig. 5

Minor comments:

- Line 309: host (maternal and fetal) factors should also be mentioned here, as I imagine they also influence infection severity
- Line 444: the table contains all that is needed but if the sequences have a common BioProject ID it should be listed in the main text for ease of access

Lastly, the code repository is not particularly user friendly, and could use some more documentation. As far as I could tell it just contains data and a couple of scripts.

Reviewer #2:

Remarks to the Author:

In this well-written manuscript, Chaguza et al. perform GWAS on whole-genome sequences from a large set of invasive pediatric group B *Streptococcus* (GBS) isolates collected over two decades through a Dutch bacterial surveillance network.

The group elects to focus on possible GBS genetic contributors to two important variables--time of disease onset (analyzed as a categorical variable, early- vs. late-onset, and also as a continuous variable) and whether or not the isolate came from the bloodstream or the CNS. These factors are clinically important and the group's decision to focus on them makes sense. The overall design of the study and its execution are sound.

The major findings are that:

- 1) The distribution of capsular serotypes, MLST groups, and clonal complexes, follow global trends and track well with other epidemiological research. This provides external validation to the dataset.
- 2) GBS genes within and immediately surrounding the capsular polysaccharide biosynthesis locus show sequence variance statistically related to both time-of-disease and CNS penetration. This makes sense, as the GBS capsule is the major interface between the bacterium and the host. The authors identify several genes-- including those encoding an arsenate reductase, acetyl esterase, a chloride channel, and a hypothetical protein--that have not been the focus of previous focused study, and which offer attractive targets for future mechanistic investigations.

The major value of this study is the large, well-characterized, and uniformly analyzed GBS collection upon which it is based. The figures, tables, captions, supplemental data, and main text are all well-composed and easy to follow. In addition to the findings reported in the article, the sequencing data made publicly available through this study will be a valuable resource to the GBS research community moving forward.

Critiques (minor):

I could not tell from the text or supplemental tables whether any of the isolates came from the same patient (either simultaneously obtained CNS and bacteremia isolates or repeat positive blood or CSF cultures from the same patient). Please clarify.

Line 116: Should read (Fig. 1 b-d). Current draft missing the numeral 1.

Thomas Hooven, MD

REVIEWER COMMENTS

Reviewer #1 (Remarks to the Author):

In this manuscript, Chaguza et al. use statistical genetics to test if neonatal infections caused by Group B Streptococcus (GBS) are modulated by bacterial genetic variants. They use a large longitudinal cohort of >1'300 GBS strains collected over 30 years in the Netherlands, for which the date of onset of the disease was recorded, as well as the isolation source. Using various classes of genetic variants and association methods, they find a few variants associated with either disease onset time and with isolation from cerebrospinal fluid.

I found the analysis to be appropriate and the results to be of interest. The heritability estimates for the target phenotypes (both related to infection severity) seem to be comparable with other recent studies. I have some comments regarding the data and the methods applied, and a few major annoyances with the presentation of the data.

Thank you for the excellent assessment.

Major comments

Are there any known biases in this dataset? As strains and data has been collected for over 30 years, there might have been changes in protocols and similar changes. If that was the case it might be worth it to add a covariate to the analysis.

Related to this, the authors report that the number of meningeal infections may be underestimated due to lack of lumbar sampling. Is there any way that this bias could be estimated by the authors and how this would affect their results?

We agree that over such a long period there may be some changes to national guidelines, policies, and the protocol. For example, around 1998, a nationwide guideline that recommends intrapartum antibiotic prophylaxis to women with risk factors of a newborn with the early-onset disease was implemented; however, based on our experience, this is unlikely to introduce bias in the dataset. In a previous publication (PubMed: 25444407), we used the concurrent *E. coli* collection to make the case that there had been no apparent shift in laboratory surveillance practice. We have also compared meningococcal submissions (all ages blood and CSF) to the reference laboratory to another mandatory notification system (see fig 1 in PubMed: 25104306) and found a similar pattern over time. In [https://doi.org/10.1016/S2666-5247\(20\)30192-0](https://doi.org/10.1016/S2666-5247(20)30192-0) (GBS CSF isolates of all ages and neonatal blood isolates), we reported that ≈90% of isolates of all patients with bacterial meningitis were sent to the National Reference Laboratory of Bacterial Meningitis (NRLBM); So although the surveillance data will be an underestimation of true incidence rates because most, but not all isolates are collected, this has been stable over time.

We agree that there is likely some underestimation of meningitis cases. Some patients will have been too sick to have a lumbar puncture performed, or conversely, some newborns will have had no signs of meningitis and antibiotics could have been started after blood culture only. In some but not all meningitis cases, a postponed lumbar puncture could have been performed with some of these CSF cultures showing no growth due to the commencement of the antibiotic treatment. However, we lack extensive clinical data to identify these cases. This means that while the CSF positive cases will be from meningitis cases, some blood culture isolates will have been from patients with unidentified meningitis. I think this misclassification would lead to the CSF and blood isolates groups being more alike than we thought and would reduce our power to identify differences between the groups, but it's unlikely to lead to false positives. Extensive clinical background information was not available, restricting the adjustments in the GWAS analyses.

We have now updated the discussion to highlight some limitations of the study.

If I understood correctly, the population structure correction was performed using a kinship matrix derived from the core genome phylogeny. Has it been corrected for the presence of recombination? Alternatively, a covariance matrix from the unitigs presence/absence matrix could be used, as that should be able to account for core and accessory genome population structure.

The population structure was corrected using a kinship matrix (or genetic relatedness matrix) generated using the SNPs inferred from the whole genome alignment of the GBS isolates created using snippy (<https://github.com/tseemann/snippy>). Indeed, a covariance matrix based on the unitigs presence/absence matrix could also be used to control the population structure. However, from our initial analysis, the results based on the kinship matrices from the SNPs and unitigs were similar; therefore, we decided to use the SNP distances. Since we analysed isolates from different lineages, we did not remove recombination events using Gubbins. In our experience, removing recombination makes sense when studying a highly clonal bacterial population, for example, isolates from a single lineage or clade, but not the entire species, as signatures for recombination from divergence through other processes are indistinguishable for highly divergent lineages.

Have the authors tried to run a burden test to see if any rare variants are associated with infection severity? Their sample size could be sufficient to see some associations.

Thank you for the suggestion. We considered doing the burden test, but we decided that it is beyond our study's scope and capability at this point. However, since we are sequencing additional GBS isolates from the same and other study settings through the GBS Juno Project (<https://www.gbsgen.net/>), we plan to do this analysis in follow-up studies with a larger number of isolates.

Can the association with a linear mixed model be run with a continuous phenotype that is not normally distributed? If not, the authors could look into approaches such as warped LMM (doi: 10.1038/ncomms5890). Judging by Supp. Fig. 7 the onset time is indeed not normally distributed.

Thank you for the suggestion. We agree that the disease onset time is not normally distributed. We have now repeated the GWAS of the disease onset time as a continuous phenotype using the rank-based inverse normal transformation disease onset time (Supplementary Fig. 7). We have updated the methods, results, and figure captions to describe that we used the transformed disease onset time as a continuous phenotype in the GWAS.

While the data and methodology are overall sound, the manuscript has several sloppy errors, which one would expect would have been caught by at least one of the 7 authors of this manuscript in the last round of editing before submission. They made reading this manuscript very frustrating. These would necessarily need to be corrected, and I would suggest the authors to double check everything to make sure none of the results, figures, and tables contain any other error. Here's a list of the errors I could spot:

Thank you for spotting the errors. We have made changes to correct the errors and checked the rest of the manuscript for any other errors.

- Line 105: Fig. 2b does not exist (that figure has a single panel), so I'd imagine the authors are referring to Supp. Fig. 2b?

We have now corrected the reference for Fig. 2b to Fig. 1b.

- Line 117, the figure number is missing

We have added the figure number (Fig. 1b-d).

- Line 126: Figure 3 is referenced before Figure 2, which is actually never referenced in the whole manuscript

We have reference Figure 2 before Figure 3.

- Supplementary Figure 2d does not make much sense, as the color and columns are the same variable?

We have now updated the order in which we referenced Figure 2 and Figure 3. As suggested, we have referenced Figure 2 before Figure 3.

- The caption for Figure 4b is incorrect, as the phenotype should be binary here, and not continuous

We previously swapped panels b and c when editing the figure, but we forgot to change the labels. We have now corrected the figure legend as suggested.

- Supplementary Figure 6 contains "volcano" and not "violin" plots

We have corrected the terminology from violin to volcano plot.

- Line 177: this sentence lacks a verb

We have now added a verb "**were**" to the sentence as suggested.

- Figure 5: the caption contains panels a to f but in the actual figure one panel identifier is missing (probably c is skipped). Also, what do the colors in this panel mean?

We have now updated the panel labels in Figure 5.

- I could not understand what is the difference between Supp. Fig. 4 and Supp. Fig. 5? The plots look slightly different, but the caption is the same

Supplementary Fig. 4 shows assembly statistics for the GBS isolates causing early-onset disease (EOD) and late-onset disease (LOD) while Supplementary Fig. 5 shows the assembly statistics for the GBS isolates sampled from the central nervous system (CNS) and non-CNS tissues. We have updated the caption for Supplementary Fig. 4 to indicate that it shows information for the EOD and LOD isolates.

- Supp. Fig. 8 lacks a legend for the colors

We agreed that the figure lacked a legend for the colours. However, we have now excluded this figure as it did not add any value to the manuscript's results section.

- Supp. Fig. 10: same as Supp. Fig. 6

We agreed that the figure lacked a legend for the colours, but we have excluded this figure as it did not add any value to the manuscript results.

- Supp. Fig. 11: same as Fig. 5

We agreed that the figure lacked a legend for the colours, but we have excluded this figure as it did not add any value to the manuscript's results section.

Minor comments:

- Line 309: host (maternal and fetal) factors should also be mentioned here, as I imagine they also influence infection severity

Indeed, we have updated the sentence to mention that the maternal and host factors also influence neonatal disease. Here is the revised sentence "Altogether, these findings suggest that although pathogen genetics **partly** influence the variability in the onset of acute invasive GBS disease and tissue invasion in the neonates and infants, **the maternal and foetal host factors**, host-pathogen interactions, and the environmental factors contribute to a more considerable extent to the GBS disease pathogenesis."

- Line 444: the table contains all that is needed but if the sequences have a common BioProject ID it should be listed in the main text for ease of access

We have now added the study accession number to the data availability statement. Here is the revised text "**The sequence reads for the isolates used in this study were deposited in the European Nucleotide Archive under study accession PRJEB14124.** The accession numbers and information **for individual isolates** are provided in Supplementary Data 1."

Lastly, the code repository is not particularly user friendly, and could use some more documentation. As far as I could tell it just contains data and a couple of scripts.

We have now updated the README file on GitHub to provide detailed information on the files and scripts included in the repository (https://github.com/ChrispinChaguza/GBS_Study_NL).

Reviewer #2 (Remarks to the Author):

In this well-written manuscript, Chaguza et al. perform GWAS on whole-genome sequences from a large set of invasive pediatric group B Streptococcus (GBS) isolates collected over two decades through a Dutch bacterial surveillance network.

The group elects to focus on possible GBS genetic contributors to two important variables--time of disease onset (analyzed as a categorical variable, early- vs. late-onset, and also as a continuous variable) and whether or not the isolate came from the bloodstream or the CNS. These factors are clinically important and the group's decision to focus on them makes sense. The overall design of the study and its execution are sound.

The major findings are that:

1) The distribution of capsular serotypes, MLST groups, and clonal complexes, follow global trends and track well with other epidemiological research. This provides external validation to the dataset.

2) GBS genes within and immediately surrounding the capsular polysaccharide biosynthesis locus show sequence variance statistically related to both time-of-disease and CNS penetration. This makes sense, as the GBS capsule is the major interface between the bacterium and the host. The authors identify several genes-- including those encoding an arsenate reductase, acetyl esterase, a chloride channel, and a hypothetical protein--that have not been the focus of previous focused study, and which offer attractive targets for future mechanistic investigations.

The major value of this study is the large, well-characterized, and uniformly analyzed GBS collection upon which it is based. The figures, tables, captions, supplemental data, and main text are all well-composed and easy to follow. In addition to the findings reported in the article, the sequencing data made publicly available through this study will be a valuable resource to the GBS research community moving forward.

Thank you for the excellent assessment.

Critiques (minor):

I could not tell from the text or supplemental tables whether any of the isolates came from the same patient (either simultaneously obtained CNS and bacteremia isolates or repeat positive blood or CSF cultures from the same patient). Please clarify.

Thank you for the suggestion. The isolates were from unique patients. When two isolates (one blood and one CSF) from one patient were stored in the Reference laboratory, the CSF isolate was selected.

Line 116: Should read (Fig. 1 b-d). Current draft missing the numeral 1.

We have updated the reference for Figure 1 from (Fig. b-d) to (Fig. 1b-d).

Thomas Hooven, MD

Reviewers' Comments:

Reviewer #1:

Remarks to the Author:

I thank the authors for reviewing their manuscript in response to my comments. Overall all comments have been addressed, with three missing points:

- the transformation of the continuous phenotype is not mentioned in the Materials and methods, and I think that it should be, at least to identify which software package (if any) has been used for this task

- My comment on Supplementary Figure 2d has not been addressed; I'm still not sure if that panel makes sense at all, and in the response to reviews Figure 2 was mentioned instead of Supp. Figure 2

- I could believe that population structure correction from SNPs is similar to the one made from the unitigs, but perhaps it could be shown quantitatively (e.g. via a scatterplot), at least as part of the peer review process

Thanks for the interesting paper and work

REVIEWER COMMENTS

Reviewer #1 (Remarks to the Author):

I thank the authors for reviewing their manuscript in response to my comments. Overall all comments have been addressed, with three missing points:

- the transformation of the continuous phenotype is not mentioned in the Materials and methods, and I think that it should be, at least to identify which software package (if any) has been used for this task

Response: Thank you for the comment. We agree with the reviewer, and we have now mentioned the phenotype transformation in the materials and methods section and provided an R code used to transform the continuous phenotype in the GitHub repository (https://github.com/ChrispinChaguza/GBS_Study_NL).

- My comment on Supplementary Figure 2d has not been addressed; I'm still not sure if that panel makes sense at all, and in the response to reviews Figure 2 was mentioned instead of Supp. Figure 2

Response: We had intended to remove Supplementary Figure 2d as previously suggested. We agree that this panel did not add anything or make sense; therefore, we have removed it from the figure and revised the figure legend.

- I could believe that population structure correction from SNPs is similar to the one made from the unitigs, but perhaps it could be shown quantitatively (e.g., via a scatterplot), at least as part of the peer review process.

Response: Thank you for the suggestion. We agree that correcting the population structure with unitigs could be similar to or better than using SNPs. To address the reviewer's concern, we have repeated the GWAS analyses correcting the population structure using unitigs instead of SNPs as suggested and updated the relevant text in the results section, figures, and supplementary material. The results are consistent with those found before, but we also found additional statistically significant and suggestive hits.

- Thanks for the interesting paper and work.

Response: Thank you for your insightful comments, which have improved the manuscript.

Reviewers' Comments:

Reviewer #1:

Remarks to the Author:

Thanks for this final round of corrections. I think that all my concerns have been addressed.